# Digital Typhoon: Long-term Satellite Image Dataset for the Spatio-Temporal Modeling of Tropical Cyclones

**Asanobu Kitamoto**[1,2]    **Jared Hwang**[3,1]    **Bastien Vuillod**[4,1]    **Lucas Gautier**[5,1]

**Yingtao Tian**[6]    **Tarin Clanuwat**[6]

[1] National Institute of Informatics, Japan
[2] Typhoon Science and Technology Research Center, Yokohama National University, Japan
[3] University of Southern California, USA
[4] Grenoble-INP, Ensimag, France
[5] Université Clermont Auvergne, ISIMA, France
[6] Google DeepMind

## Abstract

This paper presents the official release of the Digital Typhoon dataset, the longest typhoon satellite image dataset for 40+ years aimed at benchmarking machine learning models for long-term spatio-temporal data. To build the dataset, we developed a workflow to create an infrared typhoon-centered image for cropping using Lambert azimuthal equal-area projection referring to the best track data. We also address data quality issues such as inter-satellite calibration to create a homogeneous dataset. To take advantage of the dataset, we organized machine learning tasks by the types and targets of inference, with other tasks for meteorological analysis, societal impact, and climate change. The benchmarking results on the analysis, forecasting, and reanalysis for the intensity suggest that the dataset is challenging for recent deep learning models, due to many choices that affect the performance of various models. This dataset reduces the barrier for machine learning researchers to meet large-scale real-world events called tropical cyclones and develop machine learning models that may contribute to advancing scientific knowledge on tropical cyclones as well as solving societal and sustainability issues such as disaster reduction and climate change. The dataset is publicly available at `http://agora.ex.nii.ac.jp/digital-typhoon/dataset/` and `https://github.com/kitamoto-lab/digital-typhoon/`.

## 1   Introduction

Tropical cyclones, also known as typhoons and hurricanes in certain regions, have been the critical target of research due to their substantial societal impact [11]. To reduce the impact of tropical cyclones, the meteorological community, along with other earth science communities, has been developing both a theoretical and an empirical understanding of tropical cyclones through efforts such as advancing satellite remote sensing and atmospheric simulation models of higher spatial, temporal, and spectral resolutions for better analysis and forecasting.

Meteorologists have also developed an empirical method, known as the Dvorak technique [10, 54], to estimate the intensity of a tropical cyclone based on time-series observation data collected from worldwide ground sensor networks, meteorological satellites, and reconnaissance flights. This technique consists of a manual procedure to estimate tropical cyclone intensity based on the cloud

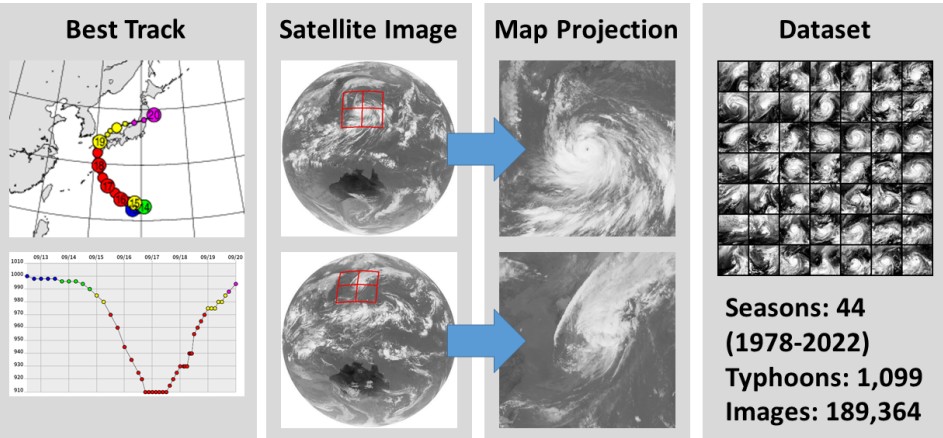

Figure 1: Overview of the Digital Typhoon dataset.

patterns of satellite images and a temporal model for intensity change. The method was originally developed in the United States in the 1970s and later adopted by meteorological agencies worldwide to become the standard procedure. However, experts are aware of its heuristic and subjective nature, as it relies on empirical, rather than theoretical, human interpretation of observation data. Solutions to this problem include more objective and automated versions of the Dvorak technique[39, 38] and a citizen science project to take advantage of collective intelligence [15].

It is clear that the Dvorak technique naturally fits into the machine learning framework by using images as input and intensity values as output. Hence there is a growing interest in both the machine learning community [42, 8, 36, 33] and the meteorology community [18, 5] to take advantage of the big data of tropical cyclones for developing data-driven approaches. One of the authors, Asanobu Kitamoto, started the Digital Typhoon project in 1999 with the aim of applying machine learning to typhoon analysis and forecasting [21, 26]. The first step was to develop a homogeneous satellite image dataset for machine learning as in Figure 1. The second step was to apply machine learning algorithms available at the time, such as SVM [24], Generative Topographic Mapping [22], and content-based image retrieval[23], which is later evolved into deep learning-based models for classification and regression tasks [46, 41], combined with fisheye preprocessing [16]. The third step was to release the website "Digital Typhoon" in 2003 for browsing and searching datasets [25]. The remaining problem was the lack of public datasets for machine learning. There have been attempts to download the dataset via scraping of the website (e.g. [53]), but the dataset created in this way is of lower quality.

Here we introduce the Digital Typhoon dataset, the *longest* typhoon satellite image dataset. This dataset reduces the burden of researchers to start machine learning on tropical cyclones without solid domain knowledge of meteorology and satellite remote sensing. We also illustrate the variety of tasks so that researchers can concentrate on building and evaluating machine-learning models.

## 2 Related Work

### 2.1 Track Datasets

The track data includes the 'annotation' of tropical cyclones, such as location, intensity, and wind circles, based on the interpretation of meteorological experts following the established procedure (e.g. Dvorak Technique). The best estimate, obtained from a retrospective analysis after collecting all the information from the start to the end of life, is called the best track dataset.

The Digital Typhoon dataset targets the Western North Pacific basin, and the Japan Meteorological Agency (JMA) is designated as the regional center to maintain the best track dataset. Globally, the International Best Track Archive for Climate Stewardship (IBTrACS) [28] collects the best track from meteorological agencies worldwide and creates a comprehensive track dataset since 1842.

IBTrACS shows an interesting variation of the best track; namely the location and intensity of the same tropical cyclone show discrepancies across meteorological agencies [48]. This fact suggests

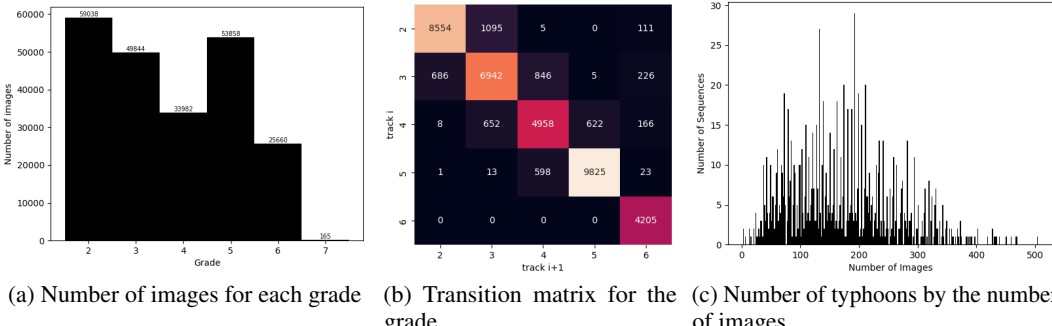

(a) Number of images for each grade  (b) Transition matrix for the grade  (c) Number of typhoons by the number of images

Figure 2: Visualization of statistics of the Digital Typhoon dataset.

that the interpretation of the observation data is not unique, or not the *ground truth* in a strict sense. Nonetheless, we regard the best track as the ground truth for most machine learning tasks, because it is the best estimate available. In a reanalysis task, however, we could critically evaluate the quality of the best track [17].

## 2.2 Image Datasets

The image dataset has information about the spatial distribution of physical properties such as cloud patterns as grid data. The observation dataset [27, 30] is derived from sensor observation that measures the physical properties of the atmosphere, while the simulation dataset, both typhoon-related [37] and the global atmosphere [44, 1, 3, 43], is generated as the representation of the atmosphere in a simulation model. Observation datasets and simulation datasets are linked through data assimilation, which is a statistical method to integrate observation datasets into a simulation model.

The Digital Typhoon dataset is an observation dataset, and it offers a richer detail of tropical cyclones with higher temporal and spatial resolutions than the simulation dataset. In addition, data quality issues in the observation dataset, such as sensor noise, missing data, and long-term sensor calibration, are handled properly so that machine learning models are not significantly affected by those issues.

## 3 Digital Typhoon Dataset

### 3.1 Dataset Overview

The Digital Typhoon dataset is created from the comprehensive satellite image archive of the Japanese geostationary satellite series, Himawari, from Himawari-1 to Himawari-9. Although those images are not copyrighted, some data are not accessible for free, and old satellite images have old formats for which open-source parsers are difficult to find. Hence we developed our own parsers for all generations of satellites, and the workflow to create typhoon-centered images by referring to the best track, as shown in Figure 1.

Using this workflow, we created the Digital Typhoon dataset by integrating metadata and images. The metadata contains hourly best-track data with additional information about the file name and each image's quality. The formatting of the best track data aligns with the original best track data sourced from the JMA. On the other hand, the images feature a 2D array of brightness temperatures around the typhoon's center, formatted in HDF5.

As a result, the dataset comprises a total of 1,099 typhoons and 189,364 images. Figure 2 visualizes some of the statistics of the dataset. It is a complete record of typhoons occurring in the Western North Pacific region (ranging from 100 to 180 degrees east of the northern hemisphere), from the 1978 season through the 2022 season, with missing typhoons in 1979 and 1980 due to the unavailability of satellite data. The length of the dataset, spanning 44 typhoon seasons (years), is the *longest* typhoon image dataset. We call it the longest dataset because Japanese geostationary satellite images for typhoons before 1978 were lost forever, and our dataset went back to the oldest satellite image preserved. Hence it provides a unique opportunity to challenge long-term datasets.

Table 1: Comparison between the Digital Typhoon dataset and the HURSAT dataset.

|  | Digital Typhoon dataset | HURSAT dataset |
| --- | --- | --- |
| Temporal coverage | 1978-2022 (present) | 1978-2015 |
| Temporal resolution | one hour | three hours |
| Target satellites | Himawari | SMS, GOES, Meteosat, Himawari, FY2 |
| Spatial coverage | Western North Pacific basin | All basins (Global) |
| Spatial resolution | 5km | 8km |
| Image coverage | 512×512 pixels (1250km from the center) | 301×301 pixels (1100km from the center) |
| Spectral coverage | infrared (others on the Website) | visible, infrared, water vapor, near IR, split window |
| Map projection | Azimuthal equal-area projection | Equirectangular projection |
| Calibration | Recalibration | ISCCP |
| Data format | HDF5 | NetCDF |
| Best track | Japan Meteorological Agency | IBTrACS |
| Dataset browsing | Digital Typhoon website | Download only |

The Digital Typhoon dataset can reduce the burden of machine learning researchers to study tropical cyclones. First, it opens up access to tropical cyclone data processed from long-term satellite data. Second, it offers a homogeneous dataset created by the image processing workflow based on expertise in meteorology and satellite remote sensing. Third, massive computations to process hundreds of terabytes of original satellite data to create a machine-learning dataset are not necessary. A comprehensive explanation of the workflow for the creation of the dataset is provided in the Appendix.

The Digital Typhoon dataset is available at the official page `http://agora.ex.nii.ac.jp/digital-typhoon/dataset/` with an open data license, namely the Creative Commons Attribution 4.0 International (CC BY 4.0) License.

### 3.2 Comparison with the HURSAT Dataset

Among satellite image datasets of tropical cyclones, Hurricane Satellite Data (HURSAT) dataset [27, 30] from The National Oceanic and Atmospheric Administration (NOAA) is the most notable dataset in size and coverage. Table 1 provides a comparative summary of the Digital Typhoon and HURSAT datasets. There are distinct variations between the two as enumerated below.

**Temporal coverage** The Digital Typhoon dataset is continually updated, and is the *longest* tropical cyclone image dataset worldwide. On the other hand, the HURSAT dataset stopped updating in 2015.

**Temporal resolution** The Digital Typhoon dataset has a temporal resolution of one hour which is higher than the HURSAT dataset's three-hour resolution. A high-frequency change such as rapid intensification is more sensitive to temporal resolution.

**Spatial coverage** The Digital Typhoon dataset specifically targets the Western North Pacific basin, whereas the HURSAT dataset encompasses all basins.

**Spatial resolution** The Digital Typhoon dataset possesses a spatial resolution of approximately 5km, superior to the HURSAT dataset's roughly 8km (0.07 degree). A small-scale structure such as the eye of a tropical cyclone is more sensitive to spatial resolution.

**Spectral coverage** The Digital Typhoon dataset incorporates the infrared (IR) channel, while the HURSAT dataset has more channels. It should be noted, however, that the Digital Typhoon *website* has the same spectral coverage, and the *dataset* can be easily extended to cover these channels.

**Map projection** The Digital Typhoon dataset utilizes the Lambert azimuthal equal-area projection, maintaining the spherical shape of the tropical cyclone, while the HURSAT dataset employs the

equirectangular (lat/long) grid, causing shape distortion in higher latitude or peripheral areas. Figure 1 shows an example of distortion when a typhoon is observed in the north.

**Dataset browsing**    The Digital Typhoon *dataset* can be browsed via the Digital Typhoon *website*, which offers additional data. In contrast, the HURSAT dataset is solely available for download.

### 3.3   Design Choices

The dataset has several design choices, such as spectral coverage, spatial resolution, temporal resolution, and spatial coverage. In the following, we explain the reasons behind our choices.

**Spectral coverage**    The current dataset includes only the Infrared channel (IR1) (wavelength of around $11\mu$m) but does not include any other channels available on the Digital Typhoon website. The following is the summary of the availability of each channel on the website.

- IR1 (infrared): the data has been available since the beginning (1978).
- VIS (visible): the data has been available since the beginning (1978), but images from early satellites were too noisy and not appropriate for a machine learning dataset. In addition, the visible channel is meaningful only during the daytime.
- IR2 (infrared) and WV (water vapor): the data has been available since 1995 (Himawari-5).
- NIR (near infrared) and other channels: the data has been available since 2005 (2nd generation) or 2015 (3rd generation).

As summarized, the IR1 is the only channel that is the longest and with fewer data quality issues, and this is the reason we included only the IR1 channel in our first version of the dataset. Future inclusion of multispectral data may offer additional tasks such as multispectral classification and regression.

**Spatial resolution**    The spatial resolution of about 5km per pixel reflects the spatial resolution of the IR1 channel for the first-generation satellites from Himawari-1 to Himawari-5. This resolution has improved to 4km for the second generation and 2km for the third generation. In spite of these progresses in technology, we chose a 5km resolution because it is the best choice to create a long-term homogeneous dataset. An interesting task in the future is to transfer a machine-learning model from long-term lower-resolution datasets to short-term higher-resolution datasets so that we can take advantage of recent technology for better forecasting.

**Temporal resolution**    The temporal resolution of one hour reflects the temporal resolution of one hour for some of the first-generation satellites after Himawari-3. From Himawari-1 to Himawari-2, the temporal resolution was more than one hour, or typically every three hours. For this reason, the data before 1987 has many missing data points as an hourly dataset. This resolution has improved to 30 minutes for the second generation and 10 minutes for the third generation. In spite of these progresses in technology, we chose one hour because it is a representative interval for many types of meteorological observations.

**Spatial coverage**    The current dataset only covers the Western North Pacific basin in the northern hemisphere (NH), but the Digital Typhoon website offers the same types of images for the southern hemisphere (SH) in the Australian basin using the best track from the Bureau of Meteorology, Australia. Here an interesting question is how a model trained in NH can be transferred to SH. From a meteorological point of view, tropical cyclones in various basins are considered the same meteorological phenomena, so theoretically, the dataset can be created similarly, and machine learning results are transferable. However, we also need to consider many details that may have an impact on the actual results, such as different quality of the best track data, and different sensor characteristics and calibration methods for different satellites. A future version of our datasets and benchmarks may address these issues.

## 4   Machine Learning Tasks

The Digital Typhoon dataset serves two important roles. First, it offers a practical real-world dataset and tasks for the machine learning community to explore new models and solutions. Second, it

provides a tool for meteorologists to apply data-driven approaches in studying tropical cyclones. The following is a summary of tasks in multiple dimensions. Other lists of tasks can be found in the review [7, 55, 45].

## 4.1 Types of Inference

**Analysis**  The task is to estimate current values using the current and past data. For instance, estimating the intensity of a typhoon falls into an analysis task, as it produces information about the typhoon's intensity using both current and past data. Supervised learning within this task can be further categorized into either a classification task or a regression task, contingent on whether the target variable is categorical or numerical. Additionally, unsupervised tasks can be designed for clustering or identifying typhoons with similar characteristics.

**Forecasting**  The task is to produce future predictions based on current and past data. The forecasts can be evaluated with the actual outcomes from the real event which become available over time. The forecasting task has a sub-task called nowcasting, aimed at making short-term forecasting spanning several hours using data-driven extrapolation. Note, however, that weather forecasting is theoretically constrained by the atmosphere's chaotic nature, which states that a minor difference in initial conditions can escalate over time.

We call this task 'forecasting' instead of 'prediction' because prediction is ambiguous in machine learning. In meteorology, prediction is strictly used to mean future values, but in machine learning prediction could mean the output of a machine learning model without temporal dimension. To avoid confusion across disciplines, we use forecasting throughout the paper.

**Reanalysis**  The task is to produce the best estimate given all obtainable data. This task is especially relevant to producing a uniform dataset spanning a long period of time, such as detecting trends in tropical cyclone activity to study the effects of climate change. As addressed in Section 2.1, the best track dataset may contain errors due to technological limitations or inconsistencies from different human experts. Machine learning can potentially aid in evaluating the quality of annotated data.

## 4.2 Targets of Inference

**Intensity**  The task makes inferences on the strength and size of a typhoon. The categorical grade is used to classify both the strength and type of a tropical cyclone. A classification task uses these grades as target variables. On the other hand, the intensity of tropical cyclones is measured numerically by central pressure and maximum sustained wind. An intensity regression task uses either pressure or wind as the target variable. In addition, the metadata includes the radius of the strong wind circle that represents the size of a tropical cyclone, so we can also design a regression task for size using the radius as the target variable.

**Track**  The task makes inferences on the geographical location of a typhoon. The cyclone's center, as estimated by human experts, is represented by latitude and longitude coordinates with a precision of 0.1 degrees. A regression task for predicting the typhoon's location uses these latitude and longitude coordinates as target variables.

**Formation**  The task makes inferences on the birth of a tropical cyclone, which typically occurs in tropical regions. Among the numerous cloud clusters actively evolving in tropical regions, determining which one will evolve into a tropical cyclone presents a challenging forecasting task, making it a target for machine learning applications [6].

**Transition**  The task makes inferences on the transition from a tropical cyclone to an extra-tropical cyclone, which typically occurs in mid-latitude regions. Two types of cyclones are conceptually distinct from a meteorological perspective, but as a natural phenomenon, they are continuous. The data-driven modeling of a continuous transition process connecting two discrete concepts is a machine-learning task.

Table 2: The statistics of target values.

| Target value | Range | Mean | Standard deviation |
|---|---|---|---|
| Central pressure | 870-1018 (hPa) | 983.8 | 22.5 |
| Maximum sustained wind | 35-140 (knots) | 59.2 | 19.8 |

## 4.3 Meteorological Analysis

Machine learning can also be applied to analyze meteorological events on tropical cyclones, such as rapid intensification [2], eyewall replacement [14], and overshooting cloud tops [20]. These events may be linked with the forecasting of tropical cyclones, yet their underlying mechanisms are not entirely understood. Data-driven methodologies could potentially provide insights that contribute to the development of a novel theoretical framework for understanding these phenomena.

## 4.4 Analysis for Societal Impact

The Digital Typhoon dataset represents the atmospheric observation of a tropical cyclone, but its societal impacts are measured by different sources and modalities. For example, hazards are measured by heavy rainfall or strong winds, disasters are measured by landslides and flooding, and damages are measured by human casualties and financial loss. To construct a machine learning model to analyze and forecast the societal impact, real-world datasets from many sources should be integrated with meteorological datasets. This would enable a more comprehensive understanding of the full range of impacts arising from tropical cyclones.

## 4.5 Analysis for Climate Change

Understanding how a long-term tropical cyclone activity is impacted by climate change is a crucial topic in society [32, 29, 31, 47]. Technological and methodological evolution that occurred during the 40+ years lifespan introduces many types of biases in the dataset. While certain biases may be removed by sensor calibration, others are harder to detect such as annotation errors by human experts. The reanalysis of historical data and the creation of a homogeneous dataset can contribute to advancing our knowledge of the relationship between tropical cyclones and climate change.

# 5 Benchmarks

## 5.1 Overview of Benchmarks

**Task**   Machine learning tasks can be combined to create benchmarks for machine learning. We propose three benchmarks, 1) Analysis, 2) Forecasting, and 3) Reanalysis of the intensity of typhoons. The following summarizes some of the technical choices for benchmarking.

**Data splitting**   In meteorological time series, data are auto-correlated and one has to be careful how to split the data before starting to train a model [49]. At least, a random split for the image level must not be used to avoid overestimating the performance due to data leakage in the same typhoon sequence. Our assumption is that every sequence is independent, and we do not have to consider any leakage across sequences. So, as long as each sequence is treated as atomic when splitting the dataset, there is no limitation to using the entire dataset. Hence we apply random splits to the sequence level (split-by-sequence) or the season level (split-by-season). More complex splits can be designed, such as split by satellite generations (1978-2004, 2005-2014, 2015-2022). These designed splits are especially useful for the reanalysis task in Section 5.4.

**Performance metric**   The following benchmarks evaluate the performance by the absolute error of target values because this is easier for domain experts to understand the result. However, for machine learning experts, the relative error of target values is more intuitive. Instead of showing absolute and relative errors for each benchmark, we summarize the statistics of target values so that relative errors can be roughly estimated. For example, the best result of $10.06 \pm 0.09$ hPa RMSE in Table 3 is less than one standard deviation of error.

Table 3: The result of the pressure regression task for two architectures and three types of input.

| RMSE (hPa) | Full ($512 \times 512$) | Resized ($224 \times 224$) | Cropped ($224 \times 224$) |
|---|---|---|---|
| ResNet18 | 10.51 ($\pm$0.11) | 10.47($\pm$0.20) | **10.06** ($\pm$0.09) |
| ResNet50 | 11.12 ($\pm$0.41) | 11.63 ($\pm$0.35) | 10.09 ($\pm$0.04) |

Table 4: The result of the wind regression task for two architectures and three types of input.

| RMSE (kt) | Full ($512 \times 512$) | Resized ($224 \times 224$) | Cropped ($224 \times 224$) |
|---|---|---|---|
| ResNet18 | 10.21 ($\pm$0.19) | 10.09 ($\pm$0.08) | **9.25** ($\pm$0.25) |
| ResNet50 | 10.05 ($\pm$0.26) | 10.21 ($\pm$0.14) | 9.13 ($\pm$0.11) |

**Software and hardware**    To perform the benchmarks, we developed a Python-based software library pyphoon2, downloadable from https://github.com/kitamoto-lab/digital-typhoon/. pyphoon2 comes with a data loader and components to help build machine learning pipelines. All the experiments were performed on the internal cluster with 6 GPUs consisting of NVIDIA Quadro RTX 6000, NVIDIA Quadro RTX 8000, and NVIDIA Quadro RTX A6000.

## 5.2   Analysis for the Intensity

We propose classification tasks, which take an image as input and estimate grade as output, and regression tasks, which take an image as input and estimate a pressure or wind value as output. In the JMA best track, grades 3, 4, and 5 denote a tropical cyclone, among which grade 5 is the most intense according to the maximum sustained wind. Grade 2 signifies a tropical depression, a type of cyclone weaker than a tropical cyclone. Moreover, grade 6 corresponds to an extra-tropical cyclone, a type of cyclone having a different structure from a tropical cyclone. Central pressure in hectopascal (hPa) is recorded for all grades, while the maximum sustained wind in knot (kt) is recorded only for grades 3, 4, and 5. In the following, we describe the result of the regression task, and the result of the classification task is described in the appendix.

We explored three types of comparisons. First, we compared three architectures, namely VGG [51], ResNet [13] and Vision Transformer [9]. Second, we compared models trained on 1) full-resolution images ($512 \times 512$), 2) resized images ($224 \times 224$), and 3) cropped images ($224 \times 224$). In 2), the full region of the image is resized, while in 3), the central region of the image is cropped without resizing. The latter is inspired by the Dvorak technique, which focuses on many relevant image features found around the typhoon center. Third, we compared two target values, namely pressure, and wind.

We used the TorchVision [35] ResNet18 and ResNet50 models with a learning rate (LR) of $10^{-4}$, batch size of 16, and for 50 epochs. An 80/20 train/test split by sequence was used. The ResNet18 and ResNet50 models were trained five and two times respectively. To evaluate, we measured the root mean square error (RMSE) of the prediction from ground truth and their standard deviations ($\pm$ std).

Table 3 and Table 4 summarize the results. Firstly, ResNet50 yielded similar results to ResNet18. Secondly, cropping the images around the typhoon center yielded a lower RMSE than other choices, indicating that cropping is better than resizing in preserving features around the typhoon center, or removing non-relevant features far from the center. Training a model on the full images did not perform well due to their larger number of pixels. Furthermore, Figure 3 illustrates that regression performs better for weaker typhoons, but worse for stronger typhoons.

## 5.3   Forecasting for the Intensity

Our previous work used Recurrent Neural Network (RNN) to forecast the pressure directly from images and showed comparable performance with SHIPS [41]. In this paper, we chose another approach using a convolutional LSTM [50] to predict the next $n$ image frames of a typhoon given the previous 12 image frames and analyze the pressure from the predicted image. We adapted an implementation [40], and used a 3-layer ConvLSTM with 128 hidden dimensions. Due to resource limitations, we used $128 \times 128$ downsampled images from only the first 24 hours of a given typhoon.

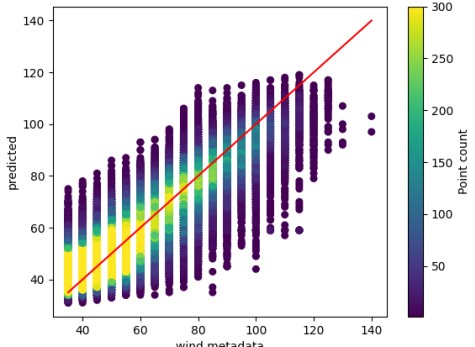

(a) Plots for wind regression by ResNet18 for cropped images.

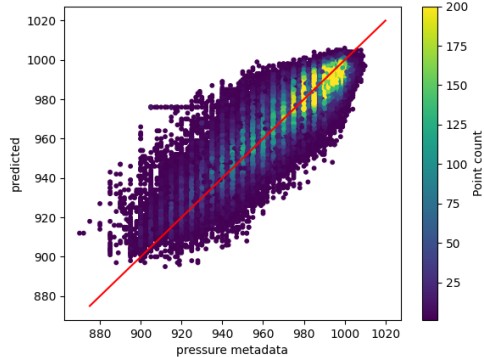

(b) Plots for pressure regression by ResNet18 for cropped images.

Figure 3: Prediction plots for pressure and wind regressions.

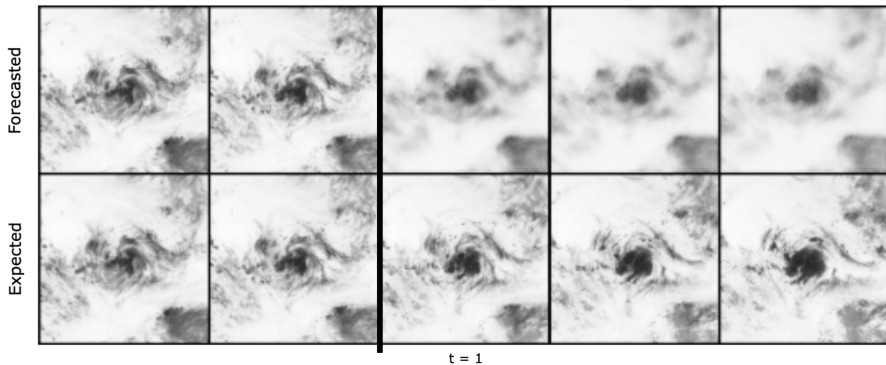

Figure 4: Results of image forecasting by ConvLSTM.

To forecast $n$ hours into the future (starting at $t = 1$), the 12 preceding frames ($t = [-11, 0]$) are passed into the model, which outputs a single image serving as its forecast at $t = 1$. Then, images from $t = [-10, 1]$, including the predicted image, are passed back into the model to get the prediction for $t = 2$. This process is repeated $n$ times to forecast $n$ hours into the future. As a result, Figure 4 shows that the first predicted frame is perceptively blurred and rapidly deteriorates as $t$ advances.

We then trained a ResNet18 model on predicted images, as well as the first 24 images of every typhoon, to predict the pressure given a $128 \times 128$ image. As a result, Table 5 shows that the model produces a larger RMSE and error as $t$ advances due to the blur of predicted images. A future adaptation may be to train both the ConvLSTM and ResNet in a black box, such that the loss is minimized by image reproduction and pressure prediction.

Both models were trained on the same 80/20 train/test split by sequence. The ConvLSTM was trained once for 230 epochs with a starting LR of $10^{-4}$, and used a CosineAnnealing scheduler [34] with 100 steps. The ResNet model used a modified first convolutional layer with a kernel and stride size of (2, 2) and (1, 1). It was trained five times with an LR of $10^{-5}$ for 34 epochs. These hyperparameters were chosen as they produced more consistent results given the smaller image sizes.

### 5.4 Reanalysis for the Intensity

The goal of this paper is to create a homogeneous long-term dataset, and the purpose of the reanalysis task is to identify biases and inconsistencies in the dataset due to factors such as technological evolution arising from satellite sensors, or methodological evolution arising from the improvement of the Dvorak technique to annotate tropical cyclones. One approach to this challenge is to design special data splits to analyze historical factors.

Table 5: Results of pressure forecasting for 12-hours by ResNet18 (values in hPa).

| $t$ | 1 | 2 | 3 | 6 | 12 |
|---|---|---|---|---|---|
| **RMSE** | $10.24 \pm 0.73$ | $10.52 \pm 0.79$ | $11.00 \pm 0.87$ | $12.10 \pm 0.85$ | $14.69 \pm 0.91$ |

Table 6: The results of regression tasks for each satellite generation (values in hPa).

| **RMSE** | Train the First | Train the Second | Train the Third |
|---|---|---|---|
| Test the First | 10.04 ($\pm 0.17$) | 9.92 ($\pm 0.09$) | 10.03 ($\pm 0.10$) |
| Test the Second | 12.80 ($\pm 0.19$) | 11.05 ($\pm 0.10$) | 11.17 ($\pm 0.10$) |
| Test the Third | 10.34 ($\pm 0.17$) | 10.03 ($\pm 0.08$) | 9.94 ($\pm 0.16$) |

Our previous work studied this task by training the model using recent data and testing on past data to analyze the trend of model performance, indicating that old satellite data may have different characteristics [41]. In this paper, we split the dataset into three buckets by satellite generations, namely the first generation (1978-2004), the second generation (2005-2014), and the third generation (2015-2022), and train and test a ResNet18 model for the regression task.

Input images were resized to $224 \times 224$ in the same way as the analysis task. 208 sequences (the size of the smallest generation) were then randomly sampled from each generation five times and split into 80/20 train/test sets. A ResNet18 model was trained on each bucket, each for 101 epochs with a batch size of 16 and a learning rate of $10^{-4}$; these parameters were chosen for their consistent results. Each of the three models was then tested on the test set of each bucket, such that a model trained on the first bucket was tested on the first, second, and third buckets.

Table 6 shows that all three models performed roughly similarly on all three buckets, and no dataset bias was immediately reflected in the quality of the models. An expansion on the reanalysis task experiment we performed is described in the Appendix.

### 5.5 Comparison with Other Approaches

Machine learning is not the only approach for data-driven analysis and forecasting of tropical cyclones. For the analysis of intensity, the Dvorak technique has been the most popular method among meteorologists. In addition, for the forecasting of intensity, computational approaches represent a typhoon in a simulation model and compute the future based on the theory of the atmosphere. This approach, however, has limitations due to spatial and temporal resolutions, and intensity forecasting is still considered a difficult challenge. Instead, meteorologists have developed empirical methods, such as SHIPS [12, 56] with linear regression on hand-crafted meteorological features, or a similar approach using XGBoost [4]. This paper focused on the comparison of machine learning models, but the real challenge for domain experts is comparing not only machine learning approaches but also computational, empirical, or manual approaches in the context of real-world solutions for tropical cyclones, such as disaster reduction. This paper is a starting point for this grand challenge.

## 6 Conclusion

We have introduced the Digital Typhoon dataset for machine learning and meteorology communities to promote data-driven research on tropical cyclones. Our dataset offers a unique opportunity to benchmark various types of machine learning models, especially spatio-temporal models for long-term time-series images. A solution is not only valuable for machine learning benchmarking but also has the potential to contribute to advancing scientific knowledge on tropical cyclones as well as solving societal and sustainability issues such as disaster reduction and climate change.

### Acknowledgments and Disclosure of Funding

Three of the authors, Jared Hwang, Bastien Vuillod, and Lucas Gautier, have been supported by the international internship program of the National Institute of Informatics.

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
