Table 7: Results of the classification task for three architectures and three types of input.

| Acc (%) | Full ($512 \times 512$) | Resized ($224 \times 224$) | Cropped ($224 \times 224$) |
|---|---|---|---|
| VGG | 70.0 ($\pm1.4$) | 67.2 ($\pm0.2$) | 69.0 ($\pm0.6$) |
| ResNet18 | 66.6 ($\pm0.7$) | 66.3 ($\pm0.9$) | 67.8 ($\pm0.2$) |
| ViT | / | 62.0 ($\pm0.2$) | 64.9 ($\pm1.2$) |

## A    Additional Benchmark Results

We already introduced benchmark results for three types of tasks in Section 5. The following are additional results of the same tasks, which were omitted due to space limitations.

### A.1    Analysis for the Intensity

In comparison to the regression task in Section 5, we summarize the result of the classification task. Here the target value is the categorical value of grades 2, 3, 4, 5, and 6. To compare the performance of the three types of architectures, namely VGG, ResNet18, and Vision Transformer (ViT), we run 2 training sessions on three types of input images. For all experiments, we used a learning rate of $10^{-4}$, batch size of 16, an 80/20 train/test split by sequence, the SGD optimizer, and 50 epochs. ViT for the full-size images was not performed due to the lack of memory.

Table 7 summarizes the result of classification in accuracy. VGG is slightly better than ResNet18, while ViT performs the worst. This result indicates that the dataset size may not be large enough for models that performed worse. However, we need to work in two directions to explore better models. First, we need to explore deeper architectures, such as ResNet50 or even deeper ResNet, to see how the depth of the network relates to the performance. Second, we need to increase the number of epochs for some models, because the best accuracy of ViT was reached during the last epochs.

Note that the result of Table 7, showing a classification accuracy of 70% at best, is likely to be an underestimate of the actual performance. This is due to the definition of grades, used as the target values of the classification task. As stated earlier in Section 5.2, grades 3, 4, and 5 denote a tropical cyclone with the ascending order of intensity, while grades 2 and 6 denote a tropical depression and an extra-tropical cyclone respectively. These categorical values, however, are continuous and ambiguous as labels for classification.

First, grades 3, 4, and 5 are continuous categories and hence the boundary between categories is not clear-cut. Considering the potential annotation errors in the best track dataset as addressed in Section 2.1, misclassification between neighboring categories should not be considered fatal mistakes. Second, grades 2 and 3 are also neighboring categories that differ only in intensity, whereas grades 3-5 and 6 differ in the structure of the cyclone. This suggests that the categories have semantics that may have different impacts on misclassification. Hence the introduction of a more realistic loss function between categorical transitions is expected to alleviate these problems.

Considering the difficulty in evaluating the classification task, we suggest that the regression task, especially on the central pressure, has a less ambiguous definition of the task. This is the reason that we introduced the result of the regression task in the main text.

### A.2    Reanalysis for the Intensity

In Section 5.4, the reanalysis task was performed on data splits by satellite generations. To observe if the dataset size has a valuable positive impact on the model performance, instead of sampling 208 sequences from each generation, we used the entire generation's sequences as the dataset to train each model (resulting in a train set size of 531 sequences, 183 sequences, and 167 sequences respectively). We then applied the same methodology used in Section 5.4 to train 3 ResNet18 models for 101 epochs with 6 training sessions.

Comparison of Table 8 with Table 6 suggests that a larger dataset size may outweigh any negative impact of inter-generational bias, as the model trained on the first bucket (the largest) performed the best.

Table 8: The results of regression tasks for each satellite generation using the entire generation (values in hPa).

| RMSE | Train the First | Train the Second | Train the Third |
|---|---|---|---|
| Test the First | 9.16 (±0.05) | 10.43 (±0.09) | 10.51 (±0.11) |
| Test the Second | 9.23 (±0.09) | 10.39 (±0.18) | 10.16 (±0.09) |
| Test the Third | 9.56 (±0.11) | 9.96 (±0.10) | 10.22 (±0.08) |

# B  Data Collection

## B.1  Image Dataset

**Accessibility**   The satellite image dataset, as introduced in Section 2.2, was built on geostationary meteorological satellites, Himawari-1 to Himawari-9 from JMA and GOES-9 from NOAA. The data was delivered through the following organizations.

1. Japan Meteorological Business Support Center (1978-1995, 2003-2022)
2. Institute of Industrial Science, The University of Tokyo (1995-2003)

Himawari satellite data is, in principle, not copyrighted, but the current situation is far from open and easy-to-download data. Some recent data is available online, but historical satellite data, especially satellite data in the original format, becomes less accessible as time goes back.

**Data format**   After obtaining the original satellite data, their processing poses another challenge due to special data formats. Himawari satellite data has three types of data formats.

1. VISSR / S-VISSR format for the first-generation satellite (Himawari 1-5, GOES 9)
2. HRIT format for the second-generation satellite (Himawari 6-7)
3. Himawari Standard format for the third-generation satellite (Himawari 8-9)

Due to the lack of standard open-source tools for parsing data formats of all generations, it is not easy for a user to create a data parser for the original satellite data. To solve this problem, we implemented our parser from scratch based on the official documentation published by JMA. We then converted the original data to a standard format for which users can find an open-source data loader library.

**Channels**   Himawari satellite data has a few channels to observe the atmosphere in different wavelengths. The visible channel ($0.6\mu$m) measures the reflection and scattering of the sunlight from the Earth, so it has advantages in observing the detailed structure of the cloud patterns. It has a higher resolution but can be used only in the daytime. The infrared channel ($11\mu$m) measures the radiation of the Earth, roughly corresponding to the temperature of the atmosphere. It has a lower resolution but can be used both day and night. Due to the continuity of the observation, we used the infrared channel to create the standard hourly satellite image dataset for typhoon images.

Note that recent satellites have more channels. Since Himawari-5, we have the water vapor (WV) channel ($6.7\mu$m) to measure the moisture in the atmosphere, and the near-infrared (NIR) channel ($3.9\mu$m) to measure the vegetation on the ground. Images of those channels are processed and provided from the Digital Typhoon website and may be included in the future release of the Digital Typhoon dataset.

## B.2  Track Dataset

The track dataset, as introduced in Section 2.1, is based on the best track data from JMA, available at `https://www.jma.go.jp/jma/jma-eng/jma-center/rsmc-hp-pub-eg/besttrack.html`. We incorporated most of the best track data, such as grade, location, central pressure, maximum sustained wind, wind circles for the storm wind (50kt) and gale wind (30kt), and the indicator of landfall or passage. The best track record is available every six hours or at shorter intervals in special cases.

Because the satellite image dataset is an hourly dataset, the best track should be interpolated for missing hours. We designed the interpolation method for each variable so that the interpolation aligns with the design of machine learning tasks that use some of those values as 'ground truth' for supervised learning.

**Location**    Location is interpolated using cubic splines [26] to create a smooth trajectory connecting best track locations.

**Grade**    The grade is persistent throughout the interpolation interval because it is a categorical value.

**Central pressure**    Central pressure is interpolated using a linear function connecting best track pressures. This is a choice to conserve maxima and minima and avoid the shooting of pressures beyond observed values. At the same time, a linear function can reflect continuous changes in the intensity of the typhoon.

**Maximum sustained wind**    The maximum sustained wind is persistent throughout the interpolation interval. The wind is a numerical value, so it matches well with the linear interpolation. However, we should consider the fact that the wind is only available for grades 3, 4, and 5, but not for grades 2 and 6, where the wind value is set to zero. Grade 2 is called a tropical depression which has a smaller maximum sustained wind than 35 knots. Because the purpose of the JMA best track is to create the record of typhoons with more than 35 knots, JMA simply does not estimate the maximum sustained wind of tropical depressions. In a similar manner, Grade 6 is called an extra-tropical cyclone, which has a different atmospheric structure from a tropical cyclone. Although the maximum sustained wind of extra-tropical cyclones may exceed 35 knots somewhere far from the center, the JMA best track does not contain the maximum sustained wind for Grade 6 for the same reason as Grade 2.

This leads to discontinuity of the wind values at the time of formation (from grade 2 to grade 3) and transition (from grade 3-5 to grade 6) of typhoons. To avoid inappropriate linear interpolation at discontinuous points, we decided to keep the value persistent. The same rule applies to wind circles.

## C    Data Processing Workflow

After data collection, the original satellite image is fed into a data processing pipeline, which has been developed by the author (Asanobu Kitamoto) since 1992. The pipeline consists of software codes written in C and Perl. They are not open-sourced due to the complex structure of software. Note that an open-source software library `pyphoon2` has been released as introduced in Section 5 to work with the Digital Typhoon dataset for machine learning tasks.

The summary of the pipeline is as follows.

1. Parse the original satellite image and create a map-projected 2D array image with digital count as pixel values (Section C.1).
2. For each pixel value, convert digital count (integer value) to brightness temperature (floating-point value). The conversion is done in two steps (Section C.2).
   (a) The first step is to convert the digital count to brightness temperature using the conversion table for each sensor.
   (b) The second step is to convert brightness temperature to calibrated brightness temperature using inter-calibration parameters from the recalibration project [52, 19].
3. The calibrated brightness temperature, including a masked pixel value, is saved in the HDF5 file (Section C.3).

In the following, we summarize the details of each step.

### C.1    Map Projection

The map projection is used to create a typhoon-centered image. When a typhoon is observed from a satellite, the shape of the typhoon is usually distorted from the satellite's viewpoint due to the curvature of the Earth's 3D surface. Here the role of map projection is two-fold. First, the map

projection creates a typhoon image where the typhoon center is located at the center of the image. Second, the map projection reduces the distortion of the typhoon by creating an image from a viewpoint above the typhoon center.

The choice of the map projection method depends on the choice of metric properties that should be preserved at the expense of others. For the Digital Typhoon dataset, we chose the Lambert azimuthal equal-area projection to preserve the area and a circular shape around the center. This is in contrast to the choice of map projection in the HURSAT dataset (Section 3.2), where their choice, equirectangular projection (lat/long grid), does not preserve any metric properties. But at least, equirectangular projection is better than a simple image cropping from the original satellite image, because the distortion of the shape is reduced during the map projection.

It is an interesting question whether we can choose the "best" map projection for a machine learning dataset. Due to the wide variety of map projections, evidenced by about 150 map projections implemented in the open-source library `proj` (as of October 2023), a comprehensive quantitative comparison of map projections under a metric is not feasible. Instead, we claim that our map projection satisfies the desirable characteristics of map projection for this dataset. For example, it is well known that Mercator projection significantly distorts the shape to the north, hence it is not an appropriate choice to focus on shape. On the other hand, the choice of azimuthal projections over other projections can be justified by the fact that the center of a tropical cyclone is a special point for its tracking. Moreover, among azimuthal projections, we can choose either equidistant or equal-area. We chose the equal-area because we assume the area of cloud pixels is a more important indicator of the intensity of tropical cyclones than the distance from the center.

Next, we consider the size of the typhoon image after map projection. Here, the size has two aspects, either the size of the scene or the size of the image. First, about the size of the scene on the original satellite image, we refer to the maximum circle of gale wind (30kt), whose diameter is 1275nm (nautical mile) or 2361km recorded by Typhoon 199713. Based on this record, we set the diameter of the scene as 2500km. Second, about the size of the image, we decided to use a square image of 512 pixels. This amounts to a resolution of 4.88km/pixel, which is close to the maximum resolution of 5km for the infrared image of old-generation satellites.

The last choice is the geometric transformation for map projection or output-to-input mapping. Some typical choices are bi-linear or bi-cubic interpolation, but they could introduce non-existent pixel values due to the mixture of observation values from multiple pixels, and conversion from digital count to brightness temperature, which will be addressed below, becomes more complicated. To avoid this problem, we chose a simpler nearest-neighbor method that preserves original pixel values. Note that some pixel values are copied more than once when the resolution of the original satellite image is not enough due to the curvature of the Earth.

### C.2   Calibration

The Digital Typhoon dataset is the collection of 40+ years of satellite data from 10 different satellites, and calibration for removing biases in each sensor is required to create a homogeneous long-term dataset. The original satellite image is recorded as the collection of the digital count, which is an integer value that records the response of the sensor. Depending on the precision of the sensor, the digital count is represented by 6 bits for a low-precision sensor and 12 bits for a high-precision sensor. From these digital counts before calibration, the following procedure is designed to produce calibrated values.

The first step is to convert the digital count to brightness temperature, which is a physical value measured in Kelvin (K) representing the temperature of the Earth such as cloud top, ground, and ocean surface. The conversion table from the digital count to the brightness temperature is provided for each sensor based on the initial calibration of the sensor.

The second step is to apply a recalibration equation to convert brightness temperature to calibrated brightness temperature. This recalibration method is effective for inter-calibration across satellite sensors to obtain even better homogeneous long-term observation data for climate change research. This calibrated brightness temperature is included in the Digital Typhoon dataset.

Note that some machine-learning papers use typhoon images that are scraped from public typhoon information websites, including the Digital Typhoon website. Using scraped images for machine

learning research, however, suffers from a number of problems. First, the JPEG format contaminates pixel values with lossy compression. Second, the pixel value of a popular image format, such as 0-255 in 8 bits, has been applied to an unknown scaling function and hence does not have a physical meaning. Third, some scraped images contain graphical elements such as the coastline and lat/long lines, and they introduce unnecessary noise. The Digital Typhoon dataset does not have those problems and machine learning researchers can focus on designing algorithms and models on a clean dataset.

### C.3 Masking

The final step of the pipeline is to save calibrated brightness temperature to a file. Because each pixel takes a floating point value, we decided to use the HDF5 (Hierarchical Data Format 5), which is a popular data format for scientific applications to deal with a multidimensional array of floating point values.

Here a problem arises with pixels that do not have observation data. This happens in the following situations.

1. A pixel is out-of-frame. It occurs when a typhoon is located at the periphery of a satellite image, and hence a typhoon-centered image overlaps with the boundary of the satellite image. It also occurs when a satellite observation is partial due to scheduled maintenance or emergency operations.

2. A pixel is contaminated by noise. It occurs in old satellites when the sensor malfunctions. These noises can be easily detected when the pixel value takes the minimum or maximum values of the range but is more difficult to detect when the noise is only visible as an irregular spatial pattern.

In HDF5, there is no standard way to represent a pixel value without observation (invalid or null values). Hence we used a temperature of 130.0, which is below the valid brightness temperature (130K or about -140 degrees Celsius). They can be used to mask invalid pixels when machine-learning models can properly treat masked pixels.

Note that we can further normalize the brightness temperature as input to machine learning models. In our benchmarking experiments, we used the standard normalization procedure to map a brightness temperature of $[170, 300]$ to a normalized value of $[0, 1]$. In this setting, the masked pixels are mapped to 0 together with other extremely cold pixels, but note that the minimum temperature of 170K rarely happens in the natural environment.

The final step is to decide if we put the image into the dataset. If an image has less than 30 percent of masked pixels, the image is included in the dataset with masked pixels. Otherwise, the image is not included and becomes missing data in the dataset. Hence the Digital Typhoon dataset consists of both perfect images without noise and imperfect images with noise or occlusion. To deal with imperfect images, pyphoon2 has a filtering option to exclude some images from the machine-learning task according to the percentage of masked pixels, and it is the user's responsibility to treat filtering appropriately for their machine-learning tasks.

## D  Dataset Organization and Updating

The directory structure of the dataset is summarized in Figure 5. The dataset consists of the image directory, containing the list of HDF5 files for hourly images grouped by each typhoon, and the metadata directory, containing one file for each typhoon with the record of best track data and the quality of the image of the corresponding observation time. At the top level, we also provide the metadata.json file, which contains information about each typhoon in the dataset, such as the season, the number of images, and the name of the typhoon.

The Digital Typhoon dataset will be updated when all the best track for a typhoon season is available, which is typically January or February of the next year in the case of the JMA best track. At the same time, the best track for some old typhoons may be updated based on the result of recent reanalysis activities. Hence a new dataset will include new data for the last season in addition to data for the past seasons with minor updates.

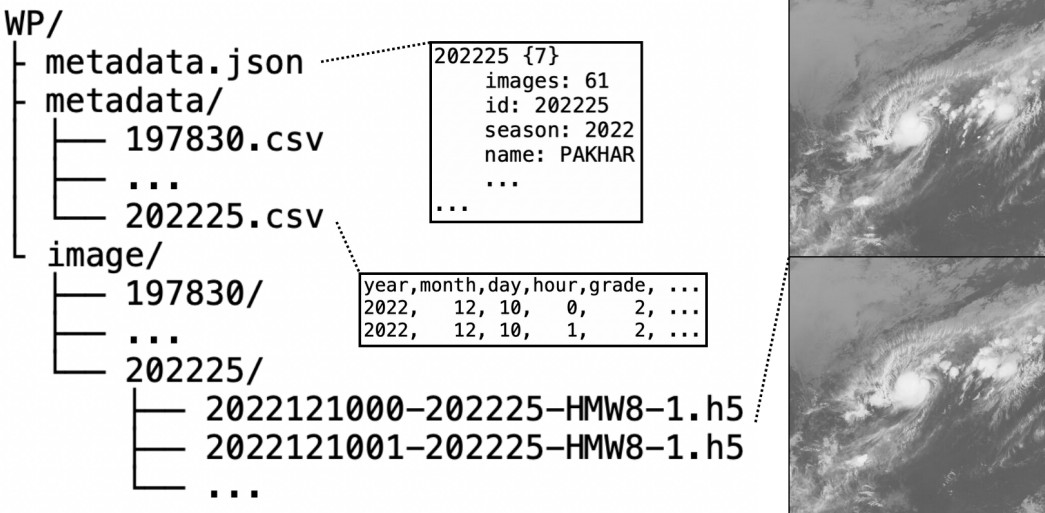

Figure 5: Directory structure of the Digital Typhoon dataset

The current dataset includes data until the 2022 season and the whole dataset is zipped into one file with the size of 54GB. In January or February 2024, we will create a dataset until the 2023 season, and replace the current ZIP file with a new one. The old data may be kept for a while for reproducible research, but roughly speaking, this is not necessary because the research is mostly reproducible by just filtering the data for the latest season. Hence it is important to record the last season of the experiments when reporting the results of experiments.

In the future, we may create related datasets to the main dataset. For example, we can release a temporary dataset for the current season, or even real-time datasets for experimenting with forecasting in real-time. In addition, we can release another dataset with higher spatial and temporal resolutions for third-generation satellites. They will be a part of the Digital Typhoon dataset as separate ZIP files so that users can choose the dataset for their needs.

## E   Responsible Use

The dataset should be used responsibly so that machine learning results do not mislead the public in terms of disaster preparedness and response. For example, in Japan, the public announcement of typhoon forecasting in real-time is only allowed for JMA, and for others, it is strictly prohibited by law. There may be similar regulations in other countries for public safety. Because typhoon forecasting is critical information for life-and-death decision-making, responsible use is required for real-time forecasting by machine learning. However, this regulation does not apply to machine learning benchmarks for historical datasets.

Another issue for responsible use is aligning with other scientific communities, such as atmosphere and climate sciences when publishing controversial results that might influence the future of human society. As discussed in Section 2.1, long-term data has many types of data quality issues such as biases that require careful modeling to produce reliable results. Machine learning research without careful consideration of those issues may arrive at misleading conclusions that raise controversies without solid scientific evidence. This problem can be alleviated by communicating with domain experts to cross-check the results from a broader perspective.

## F   Dataset Availability

The Digital Typhoon dataset will be maintained by the Digital Typhoon project at the National Institute of Informatics. This project started in 1999 and has been continuously maintaining the data processing pipeline for over 20 years. This history proves that we already established a robust system for the maintenance of the dataset.

Given that satellite images are publicly accessible observational data, we have designated the license for the Digital Typhoon dataset as the Creative Commons Attribution License (CC BY). Attribution to the dataset can be shown as follows.

> Digital Typhoon Dataset (National Institute of Informatics) doi: `https://doi.org/10.20783/DIAS.664`

Additionally, we credit two organizations as the data source. First, JMA is the organization responsible for the official data distribution. Most of our data came from JMA through JMBSC (Japan Meteorological Business Support Center). Second, satellite data between 1995 and 2003 were downloaded from the Institute of Industrial Science (IIS), The University of Tokyo, which was in charge of satellite data receiving stations and data archiving systems.

In addition, the following pages provide related information about the dataset.

**Digital Typhoon**  `http://agora.ex.nii.ac.jp/digital-typhoon/dataset/`

Digital Typhoon website offers a variety of services to explore a wide range of typhoon-related data. The official page of the dataset is available on this website.

**GitHub**  `https://github.com/kitamoto-lab/digital-typhoon/`

The GitHub page provides code to work with the dataset to train machine-learning models used for benchmarking. The code is provided with the MIT license.

**Hugging Face**  `https://huggingface.co/kitamoto-lab`

The Hugging Face page provides model weights and some codes for using them. The model is provided with the MIT license.

**DIAS**  `https://diasjp.net/`

The Digital Typhoon dataset is also available from a data repository DIAS (Data Integration and Analysis System). DIAS is a Japanese data repository for earth science and environmental datasets, hence it is a suitable place to store the dataset for long-term preservation. It also provides the dataset DOI (Digital Object Identifier) doi:10.20783/DIAS.664 or `https://doi.org/10.20783/DIAS.664` as a persistent identifier.