# OpenReview forum: "Digital Typhoon: Long-term Satellite Image Dataset for the Spatio-Temporal Modeling of Tropical Cyclones"
_NeurIPS.cc/2023/Track/Datasets_and_Benchmarks — NeurIPS 2023 Datasets and Benchmarks Spotlight_

### Official Review · Reviewer_JoXz · 2023-07-21
**A valuable extreme weather dataset**

**Rating:** 7
**Confidence:** 3
**Correctness:** I did not see any issues.
**Clarity:** Fine.

**Strengths:**

1. This is a quite useful dataset, as there are not many high-quality datasets for extreme meteorological events. Such datasets are valuable for many societal applications, and for ML research.
2. The dataset is a companion to the Digital Typhoon website, which I found useful for examining the data.
3. A companion Python package for working with the data is provided.
4. A number of benchmark results on different tasks are provided as a basis for research.

**Additional Feedback:**

The proposed update schedule suggests intermediate updates each year as typhoons occur. Per the Digital Typhoon website, there have been five typhoons so far in 2023, yet I do not see a 2023 dataset available. It would be good to discuss the expected timeframe for dataset updates.

Minor typo: Supplementary material, L737: Missing section number.

**Documentation:**

The dataset's construction and proposed maintenance is discussed adequately in the supplementary material. It is CC-BY 4.0 licensed.

One concern I have is the lack of end-to-end examples, which are valuable as a starting point or tutorial for researchers. It would be helpful if, for example, the code for the benchmark results were released (or more prominently documented).

**Ethics:**

No ethical concerns.

**Limitations:**

Limitations and tradeoffs in the dataset are discussed. One concern I have is that it was not clear from the abstract or introduction that the dataset is not global; this is not an issue, but it would be better to be clear about it early.

**Opportunities For Improvement:**

1. It would be valuable to have some additional discussion on some of the choices in the dataset construction. My preference is to make as much data as possible available, while equipping users with recommended subsets or versions for easy comparisons. In particular:
    1. How hard is it to obtain the original, processed data? This can be useful should users want to explore the effect of calibration or calibrate data differently.
    2. Why not include all available data channels (visual, IR, etc.) where available?
    3. Are higher-resolution versions of some of the data available?

2. The benchmarks could benefit from some additional baselines. In particular, what is the state-of-the-art for, e.g., forecasting? Likely this is an ensemble numerical weather model, which could provide a strong baseline where such data is available.

3. It would be beneficial for the community if tasks and areas for further research were better documented, especially for those who may not have significant meteorological expertise. What tasks would most benefit from additional research, and what level of solution quality is needed for results to be socially valuable?

**Relation To Prior Work:**

Prior work is well-discussed, and there is an extensive discussion of the HURSAT dataset in particular.

**Summary And Contributions:**

The paper introduces a dataset of satellite imagery of tropical cyclones in the western north Pacific basin, covering 1978 to the present (the paper states the dataset is continually updated). This makes easily available and unifies data that was not previously well-organized or available. The dataset is used for several benchmark tasks, along with a newly-developed Python library for handling the data.

---

> ### Author Response · Authors · 2023-08-10
>
> Thank you for review and your time reading and commenting on our work. We want to address your concerns here:
>
> 1. How hard is it to obtain the original, processed data?
>
> Recent Himawari data is easily accessible, such as open data on AWS, but as time goes back, the availability becomes worse. We need to think about several factors for availability.
>
> First, regarding accessibility to the processed data, the data is accessible from a few organizations in the world with a standard format but with lower quality than the original data. The original data is less accessible, and in particular, original data between 1978 and 1979 is much more difficult to find.
>
> Second, regarding the processability of the data, old satellite data do not come with open-source software that allow you to convert them into the dataset. In our case, we implemented our software from the specification document. Even if the original data is accessible, it is not available for analysis, including calibration.
>
> 2. Why not include all available data channels (visual, IR, etc.) where available? and are higher-resolution versions of some of the data available?
>
> These comments are discussed in responses to other reviewers. In short, the IR1 is the longest and most consistent in quality, and higher resolution data should be released as a separate dataset.
>
> 3. The benchmarks could benefit from some additional baselines
>
> We intended only to benchmark our dataset for its feasibility in machine learning, not to compare it to state-of-the-art approaches performed on other datasets. Even if the papers described the result of performance metrics, they are not directly comparable to our results due to different assumptions we made on the dataset and the tasks. We will add some descriptions of other baseline results and the comparison difficulty.
>
> 4. It would be beneficial for the community if tasks and areas for further research were better documented, especially for those who may not have significant meteorological expertise. What tasks would most benefit from additional research, and what level of solution quality is needed for results to be socially valuable?
>
> As briefly described in the paper, disaster reduction would be the most pertinent task to society. However, a solution to this topic requires additional datasets from other domains, and our dataset is mainly valuable for solving a meteorological problem such as intensity forecasting, which ultimately becomes a part of a disaster reduction solution.
>
> 5. It would be helpful if the code for the benchmark results were released (or more prominently documented).
>
> We have already made all the code used in the benchmarks available through our provided github repositories (found here https://github.com/kitamoto-lab/benchmarks/ under the releases). As for model weights, we will also release those before the final revision.
>
> 6. Per the Digital Typhoon website, there have been five typhoons so far in 2023, yet I do not see a 2023 dataset available. It would be good to discuss the expected timeframe for dataset updates.
>
> Our dataset is based on the best track data published a few months later than the real-time data. For example, the 2023 season has the best track data for only one typhoon. Due to the complicated cycle of updating datasets, it might be better to change the cycle to reduce confusion, such as periodic updates. We will improve the policy of updating the dataset and describe it in the final version.
>
> Again, thank you for your comments, and we hope we have clarified some of your concerns.

---

> > ### Comment · Reviewer_JoXz · 2023-08-13
> > **Response**
> >
> > Thank you for the detailed clarifications. These significantly address my issues, and I support acceptance.

---

### Official Review · Reviewer_SUZt · 2023-07-22
**Digital Typhoon: Long-term Satellite Image Dataset for the Spatio-Temporal Modeling of Tropical Cyclones**

**Rating:** 6
**Confidence:** 4
**Correctness:** Yes, the dataset is constructed in a …
**Clarity:** yes

**Strengths:**

- This is currently the longest temporal span typhoon satellite image dataset available.
- The authors have provided a website for convenient access to the dataset and have made the corresponding code available on GitHub.
- The paper is written in an easy-to-understand manner and is very friendly to people who have no background in meteorological knowledge.

**Additional Feedback:**

Please check the Opportunities For Improvement part

**Documentation:**

The description is relatively clear, I have raised questions in the above review.

**Opportunities For Improvement:**


- Sections 4.3, 4.4, and 4.5 could be succinctly summarized and incorporated into the Introduction.
- I find the definition and experiments of the reanalysis task confusing, particularly regarding the rationale behind dividing the dataset into three uneven buckets. I hope the authors can provide a more detailed explanation and clarification on this matter.
- In my opinion, the authors' experiments could benefit from a further comprehensive analysis of the dataset quality.
- I have downloaded your Digital Typhoon dataset, and it seems that the images are not taken at 1 hour intervals. In your naming convention of 00, 03, 06, 09, 12, 16, 18, and 21, I'm unsure whether the designation "16" is a naming error or if it indicates the usage of images specifically captured at the 16th hour.

**Relation To Prior Work:**

yes

**Summary And Contributions:**

The paper presents an open-source dataset of typhoon satellite imagery suitable for long-term spatio-temporal tasks. The dataset covers 44 years of typhoon satellite infrared images in the Western North Pacific basin, offering finer temporal resolution and spatial resolution compared to the HURSAT-B1 v06 dataset. If proven reliable, I believe this dataset could positively impact machine learning research in tropical cyclones.

---

> ### Author Response · Authors · 2023-08-10
>
> Thank you for your comments and your time reviewing our paper. We want to address your comments and concerns here:
>
> 1. I find the definition and experiments of the reanalysis task confusing, particularly regarding the rationale behind dividing the dataset into three uneven buckets.
>
> The reanalysis task is focused on examining the integrity of the dataset itself, for example, if there are biases that may have resulted from human data labeling or differences in satellite technology at different times, to list a couple of examples. In the dataset, there are three different generations of satellites, each progressively more technology capable. As such, we were interested to see if, for example, a model trained on the images from the oldest satellite would still be effective on images taken from the newest satellite (aka modern images) or if it would be more effective to use a model trained only on images from the same generation. As such, the reanalysis task is itself attempting to determine the quality and consistency of the dataset.
>
> 2. It seems that images are not taken at 1 hour intervals
>
> Thank you for reminding us of the lack of description of temporal frequency. The observations of Himawari-1 and Himawari-2 were every ~3 hours, so the one-hour interval started after 1987. We will improve the description of the dataset in the final revision.

---

### Official Review · Reviewer_zc1f · 2023-07-24
**This submission details the release of a solid dataset for tropical cyclones detection and classification**

**Rating:** 9
**Confidence:** 3

**Strengths:**

This paper seeks to bridge practitioners and researchers on the fields of deep learning, atmospheric/meteorology researchers, and remote sensing, enabling multidisciplinary studies on the field of tropical cyclones, bringing together authors with different expertise. This submission tackles this gap by providing a dataset that is ready to use (common data format, abstracting the different formats that had been used in the Himawari mission throughout the years, and with easy to ingest labels) and large enough for deep learning applications, while also incorporating some of the field-specific techniques used in tropical cyclones assessments (remote sensing TIR data, cyclones spatial features and relation to their severity classification). In terms of presentation, this paper presents the well thought reasoning used to put together this dataset, and includes a compact but complete overview of the dataset along with a comparison with one another siilar dataset in the field. It also includes the results (RMSE) of experimenting the dataset on three different deep learning models for different tasks. These experiments follow a very good description of tasks and applications for the dataset. The authors also mention different, well thought split strategies for the training and test data. In the preparation of the data, the authors discuss a very interesting point about the effects of different projections utilized when acquiring and utilizing satellite data of cyclones.





**Additional Feedback:**

No additional feedback. I have placed by comments, suggestions, and questions spread out in the other forms.

**Clarity:**

The paper is very easy to follow and it is very well written and organized. Below is a list of minor adjustments that could further improve the clarity and reduce ambiguity.

- In section 2.2, the authors refer to "the image dataset", "the observation dataset", and to "the simulation dataset". At this point in the text, it is unclear if "the" is referring to some specific datasets (in which case they should be introduced and named), or if the intention was just to identify different types of datasets.
- When reading about the Himawari satellites, it is unclear what are they observing (thermal infrared? at what wavelength?). However, that is detailed in the supplement, but it seems to be a crucial bit of information to include in the main text, or at least cross reference to the supplemental.
- Related to the point above, In Table 1, the new dataset is mentioned to include "infrared" spectral coverage. This is somewhat misleading, because "infrared" covers a broad range of the EM spectrum, and, in fact, the Himawari satellites capture IR at difference wavelengths, but only one of them is being provided through this dataset. It is later listed in the supplemental that Himawari capture NIR, MWIR and TIR, but that is ambiguous in the main document.
- Also in Table 1, when describing the HURSAT dataset, the authors specify that it includes the near infrared (NIR) band; if specifying NIR, then it calls for consistency when mentioning "infrared and near infrared". It should be specified what particular range "infrared" refers to.
- It is unclear how this dataset could be used for forecasting. It seems the reader ought to assume the formation of cyclones is also captured through the best track data. It is also unclear if the constant value assigned to wind grades (as per section B.2) could limit the ability to apply the dataset in a forecasting task. Section 5.2 illustrates a forecasting application where future images are generated. However, the authors do not show nor discuss how these future images can be used to derive an actual forecast assessment of a cyclone (e.g., its intensity). Particularly, the future images are blurry and it is not discussed if and how that would limit the detection of future cyclones or the application of those images in a regression task.
- In section 5.1, it is mentioned the limitation that hPa is available for all grades, but kt is only available in three grades. Perhaps this is obvious for a researcher in the field of tropical cyclones, but considering the broader target audience, it would be helpful to explain why there is this limitation in available data.
- Also in section 5.1, the authors conclude that cropping the images resulted in better predictions (lower RMSE) because cropping preserves better cyclone features. Couldn't it also be that there are less potential confounding factors in the input images, so not necessarily that the features are better preserved (but that there are less non-relevant features)?
- Sections 5.1 and 5.2 kick off very interesting questions on how the ML approaches compare with others. However, it turns out those questions are not addressed in the document. I found that somewhat misleading to the reader, which would expect those points to be discussed.
- In section 5.2, around line 260, the authors state that "Table 4 shows that intuitively [...]". I would argue there is no intuition being applied here, and that either an explanation is required or that simply an observation is being stated. I would suggest rephrasing this.
- In a few places throughout the document, it is written "the author [...]". However, not being an anonymous submission, it is clear that this work has various authors, such that the text should be revised to reflect that.
- Around line 737 (supplemental), there is a missing section number.



**Correctness:**

The claims in the submission seem to be correct and constructed in the a sound way. Only two points to consider:

- The authors mention they have used the "built-in torchvision [...] models". However, it is unclear where these are built-in. There is no mention/citation of using PyTorch (the library that torchvision is a part of), which could be seen as a misattribution.

- The dataset is said to include about 189k images. However, it is somewhat unclear if this is the number of images usable in practice. The authors recognize the chance for data leakage, and carefully take steps to avoid that (i.e., they split the data by sequence). So, in practice, considering the goal is to provide a dataset to be utilized in machine learning applications where data leakage is to be avoided, is the 189k number truthful? Also, it would be helpful for the reader to know the number of images utilized in the benchmark tasks, after the split by sequence, as a way to have a notion of the actual number of images usable (at least for those tasks, and with that split).

**Documentation:**

This submission is very well documented and provides all the aspects asked for. The only thing that could be added (but it is not required) is regarding the pipeline utilized to collect and process the source data. The authors argue the code utilized is very complex, hence it is not public, however one could argue that complexity alone is not an impediment for public release (e.g., the Linux kernel is quite complex and is fully open source).

**Ethics:**

No ethical concerns.

**Limitations:**

It is unclear how transferable would this dataset be across other regions. It would help the reader to know if there are no known practical differences between tropical cyclones in the Western North Pacific basin and in other basins. Similarly, it would be helpful to read about whether or not and how this dataset could be applied to tasks utilizing data from other satellites. Other limitations are mostly related with clarity, as described in the other comments. I see no potential negative societal impact.

**Opportunities For Improvement:**

When reading about the experiment detail in section 5.3, which tackles a very interesting point (data quality differences throughout the different satellites generations and their inherent technologies), it quickly came to mind that it is not possible to clearly understand the difference among the three generations if there are confounding factors such as the significant differences in the amount of training data. It is later illustrated in the supplemental material that the authors, indeed, also experimented enforcing the same number of training data for each generation. I would suggest the presentation of the experiments should be flipped: move the one in the supplemental to the main document. In any case, it would be helpful to mention in the main text that the other experiment is in the supplemental, and the discussion of the results in the main text should be made in light of both experiments.

Reading section C.3 in the supplemental, it seems that low-temperature outliers are filtered out from the dataset, as a side effect of having both the no-data/noisy pixels and the extremely cold pixels mapped out to the value 0. Generally, outliers are still part of the observed data, and as such they should be included, leaving to the researcher the option to exclude them in their applications. In the least, I would suggest this decision is clearly stated upfront in the main text.

The authors explain in the supplemental that the other bands captured by the Himawari satellites can be accessed through the website interface. Certainly, including them in the dataset would be an opportunity for improvement, in particular if features in the other bands could also be exploited to better detect and classify cyclones.

As part of the benchmarks detailed in this submission, the authors created several deep learning models, trained with the new dataset. It is unclear if those trained models are being made available through the listed repositories; if so, it would be great to have that mentioned in the document as well; otherwise, including those pretrained models in as part of the submission could be another avenue for improvement, even if only serving as baselines for future improvement

As mentioned in another comment, the authors mention the interesting and well thought implications of different projections. It would be valuable to see a quantification of the impact (e.g., RMSE) of utilizing different projections on some of the tasks. It is clear that there would be effect, but it is unclear if the differences—from a deep learning model perspective—are significant or minor.

Additional possible improvements can be achieved by addressing the comments in the other forms.

**Relation To Prior Work:**

The authors present a table comparing with another similar dataset. Additionally, the authors link back to some of their past work. However, the relation to previous contributions (datasets) is somewhat lacking.

**Summary And Contributions:**

This paper presents a new development on an existing and well established dataset on tropical cyclones (previously only available as a website content). This dataset enables future studies utilizing deep learning to detect and classify tropical cyclones. The dataset is comprised of a collection of thermal infrared (TIR) images acquired from 9 satellites (all in the Himawari mission) over the Western North Pacific basin, covering a period of 42 years. Cyclones present in those images are matched with a previously existing dataset of tropical cyclones occurrences and their classifications (best track), thus allowing to assign labels to the images. The authors included a thorough description of tasks that can benefit from this dataset, as well as some benchmarking on some of them. The manuscript text is well written and it is accompanied by informative and well designed figures.

---

> ### Author Response · Authors · 2023-08-10
>
> Thank you for your detailed review of our paper. Note that some answers are included in responses to other reviewers due to the limitation of space.
>
> 1. It seems that low-temperature outliers are filtered out from the dataset … they should be included
>
> - First, the outliers and out-of-frame pixels are counted for a projected image. If they are less than 30% of all pixels, the image is included in the dataset, assigning a pixel value of 130.0 (K) to mark them as mask pixels.
> - Second, pyphoon2 has a filtering option to exclude some images from the machine-learning task according to the percentage of mask pixels.
> - Third, pyphoon2 has a transformation to normalize pixel values to [0,1]. Here outliers are converted to the minimum value. However, the minimum value corresponds to 170K, which rarely happens in the natural environment.
>
> In short, outliers were not removed from the dataset, and it is the user's responsibility to treat them appropriately for their machine-learning tasks. In the final revision, we will make this distinction clearer.
>
> 2. It would be valuable to see a quantification of the impact of utilizing different projections on different tasks
>
> The effect of map projections is larger when the earth's curvature is apparent, such as in mid-latitude higher than 30N or peripheral areas around 100E or 180E. It suggests that the impact of map projection is significant for a transition task that must deal with extra-tropical cyclones in higher latitudes. We will show better examples to illustrate the impact of map projection.
>
> 3. It is unclear how transferable this dataset would be across other regions or other satellites
>
> From a meteorological point of view, tropical cyclones in various basins are considered the same meteorological phenomena, so theoretically, the dataset can be created similarly, and machine learning results are transferable. However, we also need to consider many details that may have an impact on the actual results, such as the quality of annotations and different calibration methods. For other satellites, the answer is similar. Theoretically, it is possible, but there will be many technical challenges in the implementation phase.
>
> 4. There is no mention of using PyTorch which could be seen as a misattribution
>
> We missed this citation and will include it in the final revision.
>
> 5. In practice, considering the goal is to provide a dataset to be utilized in machine learning applications where data leakage is to be avoided, is the 189k number truthful?
>
> We assume every sequence is independent, so we do not consider any leakage across sequences. So, as long as each sequence is treated as atomic when splitting the dataset, there is no limitation to using the entire dataset in experiments.
>
> 6. It would be helpful for the reader to know the number of images utilized in the benchmark tasks, after the split by sequence
>
> After filtering out specific samples (as described in the paper), the dataset size used in the benchmarks was 188452 images. When splitting by sequence into 80% train and 20% test, the training bucket, on average, takes 79.93% of the entire dataset. When splitting by year, the training bucket takes, on average, 78.55% of the entire dataset.
>
> 7. It is unclear how this dataset could be used for forecasting.
>
> We included the forecasting task for the intensity, but the dataset can be used for various forecasting tasks, considering the future values are ground truth. This paper proposes an algorithm for forecasting intensity in two steps: forecasting blurry images and estimating intensity from blurry images using a regression model. We will clarify the forecasting tasks in the final version.
>
> 8. Couldn't it be that [cropping the images] provides less potential confounding factors?
>
> Our idea to crop images is based on the original Dvorak technique which focuses on many features around the center. However, as you point out, it also removes non-relevant features far from the center. In the final version, we should discuss those two factors that work together.
>
> 9. It would be helpful to explain why there is a limitation [in including kt for all grades]
>
> This limitation is derived from the JMA best track data we used for the dataset.
> - Grade 2 is not a typhoon, so the JMA does not officially record the wind speed in the best track.
> - Grade 6 is an extra-tropical cyclone, which has a different structure from a tropical cyclone, hence strong winds could happen somewhere far from the center.
>
> 10. Sections 5.1 and 5.2 kick off very interesting questions on how the ML approaches compare with others. However, it turns out those questions are not addressed in the document.
>
> In this work, we focus on benchmarking our dataset on its feasibility for ML approaches and not how those specific approaches compare to existing methods outside of ML. We will move the description to other sections to clarify that our experiments do not compare ML with non-ML approaches.

---

> > ### Comment · Reviewer_zc1f · 2023-08-31
> >
> > I am very satisfied with the detailed responses the authors have provided to my queries and comments. I continue to support the acceptance of this work.

---

### Official Review · Reviewer_fsVs · 2023-07-28
**Review of Digital Typhoon Dataset**

**Rating:** 8
**Confidence:** 5
**Clarity:** Paper is well written.

**Strengths:**

- This is a very comprehensive dataset in terms of temporal coverage, and tasks that are enabled.
- The samples are well organized, and authors have provided detailed explanation how the dataset is curated.
- Users are informed how they should sample the dataset given it's inherent properties like auto-correlation.


**Additional Feedback:**

N/A

**Correctness:**

All statements and claims are correct, and well justified by the dataset. Evaluation metrics of choice are reasonable.

**Documentation:**

The dataset doesn't have a documentation in the repo. I expect authors to include that before the final release. The url is already established, and the license will be CC BY 4.0

**Ethics:**

No ethical concerns with this dataset and the paper.

**Limitations:**

Nothin beyond what authors have reported in the paper. This is a very challenging ML task, and the development of this dataset can advance ML models that can be applied for such a task that would hopefully result in identifying opportunities for further improvement of the dataset.

**Opportunities For Improvement:**

- It wasn't clear why authors selected to only include the IR band in the dataset while they claim other bands are available on their website. What's the difference between the data on the website and the benchmark? It would be beneficial to the community to include all the bands in the dataset (you can choose to store each band in a separate file to let users decide which bands they want to download/store locally).
- The paper can benefit from a section that better describes the structure and format of the dataset. This would be also needed to be included in a Documentation to accompany the dataset for the release.
- RMSE is a great metric for reporting the accuracy of regression tasks for this benchmark, but the absolute value of the RMSE has no meaning for a non-domain-expert user. e.g how good is 10 hPa of RMSE? I suggest reporting these as the percentage to the mean of the dataset.
- Section 5.3 is a great addition to the paper, but the fact that authors didn't use the same sample size for each time bucket kills the purpose of this section and it is noted at the end of the section. I ask authors to re-run the experiments for this section and select the same sample size for each bucket so results can be interpreted correctly, otherwise there is no conclusion to be derived from this section.





**Relation To Prior Work:**

This works builds on prior work from the authors, and they also compare their dataset to another existing dataset from NOAA. The comparison provides a clear picture of the pros and cons of the Digital Typhoon compared to the NOAA one and overall the new dataset is superior.

**Summary And Contributions:**

This paper introduces a new benchmark (training data, tasks and baselines) for tropical cyclone modeling in the West Pacific Ocean (hence named Typhoon). The paper builds on more than two decades of experience from the team working on using CV and ML techniques on satellite imagery for Typhoon modeling. This dataset is the longest temporal data for such a task, and it is released along with a Python package for users to interact with the data.
Multiple tasks can be implemented using such a time-series dataset, and authors implement baseline for three of them, 1) Analysis, 2) Forecasting, and 3) Reanalysis. Curating this dataset is a challenging task given the massive amount of data processing required, and authors have harmonized all the formats throughout the years to facilitate the usage of the dataset.

---

> ### Author Response · Authors · 2023-08-10
>
> Thank you for your time and effort in producing a thorough review of our work. We would like to address the points you made and how we will factor them into our final revision:
>
> 1. It wasn't clear why authors selected to only include the IR band in the dataset
>
> The downloadable dataset includes only the IR band, as it has the longest history in the dataset. The actual situation is as follows:
> - IR1 (infrared): the data has been available since 1978 and is what we have released
> - VIS (visible): the data has also been available since 1978, but the quality of Himawari-1 is very low, and the quality of early satellites was sometimes unstable. In addition, the visible bands are observable only during the daytime.
> - IR2 (infrared) and WV (water vapor): the data has been available since 1995.
> - NIR (near Infrared) and other bands: the data has been available since 2005 or 2015.
>
> As a result, in this paper, we focused on tasks that can take advantage of the long history of the IR1 and provided benchmark results on this dataset. Similarly, a multispectral dataset should be provided with associated tasks and benchmarks, but we did not address any of the issues in the paper. That is why we only included the IR1 band in our dataset.
>
> 2. Absolute value of the RMSE has no meaning for a non-domain expert user
>
> Thank you for a fair and excellent observation. We have calculated some statistics regarding the wind and pressure data that we trained our benchmarks on, which are as follows:
>
> Pressure
> - Range: 870 to 1018 HPa
> - Mean: 983.8
> - Std: 22.5
>
> Wind Speed (only the metadata of typhoons with 3 to 5)
> - Range: 35 to 140
> - Mean: 59.2
> - Std: 19.8
>
> For example, in Table 2, the best result of 10.06±0.09 hPa RMSE for the pressure task is less than one standard deviation of error. We hope this improves the interpretability of non-domain experts and will include this context in the final revision.
>
> 3. [Not using] the same sample size for each bucket kills the purpose of section [5.3]"
>
> Thank you for the suggestion, and we entirely agree with you and the sentiment that reviewer zc1f has also communicated. We want to note that we did include results for controlling the bucket size in the supplementary material. To address your comment, we will switch the two experiments such that the experiment controlling for bucket size is in the main body, and the one using different bucket sizes is placed in the supplementary material. We hope this will bolster the impact of the section. Further, we have some extended results for the experiment using controlled bucket sizes, where instead of choosing the samples once and training the model 5 times, we randomly sample that bucket size from the satellite each time we train the model. The results are within the same ballpark but may notably have more interesting trends we would like to include in the paper. The modified Table 5 would be as follows:
>
> | RMSE        	| Train the first | Train the Second | Train the Third |
> |-----------------|-----------------|------------------|-----------------|
> | Test the First  | 10.13±0.52  	| 10.45±0.35   	| 10.21±0.30  	|
> | Test the Second | 11.23±0.40  	| 10.83±0.68   	| 10.48±0.36  	|
> | Test the Third  | 10.31±0.56  	| 10.01±0.50   	| 9.82±0.38   	|
>
> The new results show the same trends as those from the original supplementary material but with a more robust experiment.
>
> Notably, the first bucket no longer shows better performance than the other two regardless of the test bucket, and the trained third bucket shows better test performance on the third bucket instead of the prior two.
>
> 4. The paper can benefit from a section that better describes the structure and format of the dataset
>
> We agree and plan to host the documentation on the website or github before full release.
>
> We hope this clears up your concerns; please let us know if you have any follow-ups or further concerns.

---

> > ### Comment · Reviewer_fsVs · 2023-08-30
> >
> > Thanks for addressing these comments. I recommend you include a brief text about the first comment (use of IR band) in the text as well.

---

### Official Review · Reviewer_Tmq2 · 2023-08-09
**A large typhoon infrared image dataset with rich annotations**

**Rating:** 7
**Confidence:** 3

**Strengths:**

S1. The dataset's extensive temporal coverage is remarkable, making it a significant contribution to the field.

S2. The integration with the Digital Typhoon website and the provision of a Python package enhance the dataset's accessibility and usability.

S3. The paper is well-structured and provides a clear overview of the dataset's creation, its potential applications, and benchmark results.

**Additional Feedback:**

The Digital Typhoon dataset is a commendable effort, bridging a gap in the availability of long-term satellite imagery datasets for tropical cyclones.

**Clarity:**

The paper is well-written, with minor areas that could benefit from further elaboration or restructuring for clarity.



**Correctness:**

The methodology and claims appear sound.



**Documentation:**

The dataset's construction is adequately detailed.



**Ethics:**

No ethical concerns were identified.



**Limitations:**

The dataset's focus on the Western North Pacific basin limits its global applicability. However, this limitation is understandable given the scope of the project.

**Opportunities For Improvement:**

O1. The proposed workflow specifically focuses on infrared images. While this is useful, it would be nice if the authors could expand this effort to cover other spectrums (e.g., visible spectrum) as well.

O2. The images have a coverage of 512×512 pixels, which is approximately 1250km from the center of the typhoon. It would be nice to also include images with higher resolution.

**Relation To Prior Work:**

The comparison with existing datasets, especially HURSAT, is well-executed, providing readers with a clear understanding of the Digital Typhoon dataset's advantages.

**Summary And Contributions:**

The paper presents the Digital Typhoon dataset, a comprehensive collection of satellite imagery spanning over four decades, focused on tropical cyclones in the Western North Pacific basin. The dataset's temporal coverage is commendable, and its integration with a Python library offers a valuable resource for researchers in the field. The authors have also provided benchmarks for various tasks, setting a foundation for future research.

---

> ### Author Response · Authors · 2023-08-10
>
> Foremost, thank you for your time in reviewing our paper and your comments. We are happy that you recognized the strength of the paper. In addition, we would like to address your comments on improving the paper.
>
> 1. The proposed workflow specifically focuses on infrared images. While this is useful, it would be nice if the authors could expand this effort to cover other spectrums (e.g., visible spectrum) as well.
>
> The current dataset includes only the IR1 band (thermal infrared around 11 micrometers) because it is the longest data with fewer data quality issues. Our benchmark only includes the result on the IR1 band for the same reason. Other bands will be released as the next step, possibly with a new benchmark on multispectral images.
>
> 2. It would be nice to also include images with higher resolution.
>
> Higher-resolution images can be obtained for only recent Himawari satellites. Because we focus on long-term data spanning more than 40 years, the spec of the dataset, such as spatial and temporal resolution, is a compromise between the past and the present. The short-term higher resolution dataset may be released in the future as a separate dataset.
>
> 3. The dataset's focus on the Western North Pacific basin limits its global applicability. However, this limitation is understandable given the scope of the project.
>
> The current dataset is limited to the Western North Pacific basin, but it can easily be extended into the Southern Hemisphere around Australia because the tropical cyclone data is already created from the same satellites. Extension to other basins is not included in our current scope because we need a new pipeline to process other satellite data.
>
> Again, thank you for your comments to improve our paper.

---

### Decision · Program_Chairs · 2023-09-22

**Decision:**

Accept (Spotlight)

**Comment:**

The reviewers are positive about this submission and are largely satisfied with how the authors have addressed their concerns. The authors are encouraged to address the outstanding suggestion to "include a brief text about the first comment (use of IR band)"